# Practical Latency Analysis of a Bluetooth 5 Decentralized IoT Opportunistic Edge Computing System for Low-Cost SBCs

**DOI:** 10.3390/s22218360

**Published:** 2022-10-31

**Authors:** Ángel Niebla-Montero, Iván Froiz-Míguez, Paula Fraga-Lamas, Tiago M. Fernández-Caramés

**Affiliations:** 1Department of Computer Engineering, Faculty of Computer Science, Universidade da Coruña, 15071 A Coruña, Spain; 2Centro de Investigación CITIC, Universidade da Coruña, 15071 A Coruña, Spain

**Keywords:** opportunistic networks, Opportunistic Edge Computing, OEC systems, opportunistic IoT, bluetooth 5, decentralized IoT

## Abstract

IoT devices can be deployed almost anywhere, but they usually need to be connected to other IoT devices, either through the Internet or local area networks. For such communications, many IoT devices make use of wireless communications, whose coverage is key: if no coverage is available, an IoT device becomes isolated. This can happen both indoors (e.g., large buildings, industrial warehouses) or outdoors (e.g., rural areas, cities). To tackle such an issue, opportunistic networks can be useful, since they use gateways to provide services to IoT devices when they are in range (i.e., IoT devices take the opportunity of having a nearby gateway to exchange data or to use a computing service). Moreover, opportunistic networks can provide Edge Computing capabilities, thus creating Opportunistic Edge Computing (OEC) systems, which deploy smart gateways able to perform certain tasks faster than a remote Cloud. This article presents a novel decentralized OEC system based on Bluetooth 5 IoT nodes whose latency is evaluated to determine the feasibility of using it in practical applications. The obtained results indicate that, for the selected scenario, the average end-to-end latency is relatively low (736 ms), but it is impacted by factors such as the location of the bootstrap node, the smart gateway hardware or the use of high-security mechanisms.

## 1. Introduction

We are heading to an era where billions of households and industrial objects will be interconnected and will be able to collaborate and to interact with each other and with their surrounding environments. In fact, some reports estimate that 75,000 million Internet of Things (IoT) devices will be in operation by 2025 [1]. Many of such IoT devices will be constrained in terms of storage, computing power, and power consumption, so they must rely on other remote devices to perform compute-intensive tasks. Moreover, smart IoT devices can be anywhere and need to be connected (e.g., to a Local Area Network (LAN) or to the Internet), but, in some areas, wireless communications coverage is not always available.

Opportunistic Edge Computing (OEC) systems have the ability to discover deployed IoT or Industrial IoT (IIoT) devices to provide them with Edge Computing services in an opportunistic way. Thus, since such IoT/IIoT devices are usually scattered throughout remote or large scenarios, their connectivity and computational tasks depend on external devices that cannot be accessed continuously. Moreover, such scattered IoT/IIoT devices usually rely on batteries to operate, so it is essential to minimize their power consumption and to optimize the resources they need to carry out their tasks.

To confront the previous challenges, this article proposes an OEC IoT architecture able to make systems independent from a continuous Internet connection in cases where communications are not possible due to the lack of wireless communications coverage or when network connectivity failures occur. Specifically, the main contributions of this work include:First, in order to establish the basics, it reviews the state of the art on OEC systems.Second, it details the design and implementation of a novel OEC system based on Bluetooth 5 and on the use of Single-Board Computers (SBCs). Such a system is decentralized (there is not a single central controller), distributed (storage is distributed among all the nodes that make up the network), scalable (the OEC system can grow by adding more nodes without affecting the service) and modular (the system is able to adapt to support different communication technologies). Moreover, the system architecture supports providing services such as Resource Sharing (to improve processing speed), Data Routing (to find the path that should take the data from one IoT node to another) or Node Discovery (to find IoT nodes easily). None of the previous characteristics have been found together in a single development described in current literature.Third, it analyzes the feasibility of the proposed OEC system through performance tests that estimate the system latency in end-to-end communications, when using a different location of the bootstrap node, for different IoT hardware and when enabling/disabling communications security mechanisms. As a result, this article provides useful guidelines for the developers of future OEC systems.Finally, all the source code used for running the experiments presented in this article is available online, thus allowing any researcher or developer to use, to adapt and to extend the developed OEC system.

The rest of this article is structured as follows. Section 2 analyzes current state of the art in relation to opportunistic Edge Computing systems. Section 3 describes the design and implementation of the proposed OEC solution. Section 4 presents the performed experiments. Finally, Section 5 summarizes the key findings, while Section 6 is dedicated to the conclusions.

## 2. State of the Art

### 2.1. Opportunistic Edge Computing Systems

OEC systems can help in challenging scenarios where IoT/IIoT devices have:No or intermittent Internet connectivity due to poor wireless coverage or to certain communications restrictions (e.g., due to their expected battery life).Limited local storage and computing power.Reduced mobility, which prevents IoT/IIoT nodes from communicating with other remote nodes and components of the IoT/IIoT architecture.

In particular, opportunistic collaboration becomes advisable in Cloud Computing environments, which are not inherently energy-efficient or secure, and have limitations when it comes to large-scale IoT implementations in terms of cost, capacity, unavailability of services or susceptibility to manipulation [2].

The technological evolution of SBCs and similar IoT devices allows them to provide a significant amount of computing power, letting them act as end nodes or gateways either in traditional Edge Computing systems [3] or in the latest Fog Computing architectures [4]. Moreover, the use of opportunistic communications by such low-power devices allows for coining the term Opportunistic Edge Computing, a concept that has similarities with other paradigms like Mobile Ad-Hoc Networks (MANETS), but it goes beyond them by providing more services than just routing capabilities. Unfortunately, different terms are used to denote similar approaches to OEC, such as Parked Vehicle Edge Computing [5], Proximal Mobile Edge Server [6], Mobile IoT [7], or Opportunistic Fog Computing [8].

Different OEC systems were already deployed in IoT fields such as wildlife monitoring [9] or smart cities [10]. However, in most of such IoT systems there is still an important requirement: they rely on the existence of an Internet connection when they need to make use of certain remote cloud services.

As an example, Figure 1 shows an OEC architecture for a smart city. In such an example, IoT smart gateways are deployed throughout the city (either at specific static spots (e.g., public buildings) or in vehicles (e.g., public transport)) to monitor, interact and provide services to vehicles, citizens and to the city infrastructure in places where no communications are available or where it is expensive to provide such communications. Thus, when an IoT node is in range of a smart gateway, the former has the opportunity of making use of the Edge Computing services provided by the latter. Such services usually involve sending data from IoT nodes to remote destinations, which is accomplished by collecting and routing them through the Smart Device and Routing layers. Then, the information can go to the Cloud, where it can be stored and processed with the help of third-party services or by combining them with information that comes from other IoT/IIoT networks.

Table 1 shows a comparison among some of the most relevant OEC solutions that can be found in the literature. The compared systems provide solutions for fields such as smart agriculture, wildlife monitoring or smart cities, and are able to make use of parked vehicles or Unmanned Aerial Vehicles (UAVs) as opportunistic Edge Computing devices.

As it can be observed in Table 1, only part of the compared opportunistic systems are decentralized (i.e., all nodes are connected without relying on one or several central servers). For instance, in [7] the authors propose a computing paradigm where mobile IoT devices are able to share unused resources (i.e., storage and computing power) in an opportunistic way. Thus, a distributed scheme is devised with the objective of allocating the available resources and for executing the provided services (this is achieved by using blockchain and smart contracts [11]). To demonstrate the feasibility of the proposed solution, the authors implemented and evaluated it on low-cost SBCs (specifically, on Raspberry Pis).

In the case of [12], the authors propose a decentralized Fog Computing architecture [13] for IoT environments that makes use of blockchain and virtualization technologies. In addition, the mentioned paper presents a prototype based on a Raspberry Pi that is used to validate the feasibility of the proposed system.

Another feature that is compared in Table 1 is modularity, which is defined as the adaptability of the proposed architecture to support different communication technologies. For example, the solutions described in [8,12,14] were devised to be modular. Specifically, in [8] the authors present a Fog Computing system whose feasibility for being used as an opportunistic Fog Computing solution is evaluated. Moreover, the authors propose a Fog Computing network architecture made up of virtual clusters, which allow for abstracting the complexity of the lower layers. Furthermore, the authors show simulation results for various scenarios in order to determine the feasibility of building an opportunistic Fog Computing network. Similarly, in [14] an edge-fog-cloud architecture is proposed to improve energy efficiency in smart agriculture systems. Such an architecture enables the data collection from various sensors to process and to analyze agricultural data that require real-time operations. Thus, real-time processing can be performed by the edge and fog layers to reduce the computational load of the cloud, which helps to improve the overall power consumption and to provide agricultural applications/services efficiently. According to the presented results, the proposed architecture reduces total energy consumption by 36% and carbon emissions by 43%.

Resource sharing is another feature compared in Table 1, since it is really important for OEC systems to improve network computational resource efficiency to run applications and to provide services close to the deployed IoT/IIoT devices. Some of the publications that deal with such a feature are [5,7]. For instance, in the case of [5] the authors present a paradigm called Parked Vehicle Edge Computing (PVEC), in which parked vehicles act as edge nodes of a vehicular network. The article describes the system architecture and a protocol that supports secure communications among the deployed network nodes. Moreover, the authors solve the resource scheduling optimization problem using the Stackelberg game approach [15].

Data routing and node discovery are also essential for OEC systems. In the case of opportunistic data routing, it is defined as the ability of the network to carry information from one node to another when the receiving node is not within the communications range of the sending node. Regarding node discovery, it is the capacity of the nodes to scan the network looking for other nodes. Among the articles that describe implementations with both features are [9,16]. For example, in [16], the authors propose a Fog Computing network based on UAVs. Thus, the system harnesses both the benefits of applying the Fog Computing principles and the mobility of UAVs, which allow the fog network to be deployed in the place where it is needed. The authors also propose a service-oriented platform where all IoT resources are regarded as a set of services to be used to develop IoT applications.

Other articles implement only one of the two previously mentioned features: a data routing solution is implemented in [17], while a node discovery mechanism is described in [6]. In the case of [17], the authors propose a distributed cloud architecture. Such an architecture is based on three technologies (Fog Computing, Software-Defined Networks (SDNs), and Blockchain) and was designed to provide high availability, real-time data delivery, high scalability, security and low latency. The results presented in the article indicate that, in comparison to traditional cloud computing infrastructure, the proposed model is more efficient for downloading data. A similar development is described in [6], where the authors evaluate and compare the performance of mobile devices when running activity detection applications that involve mobile devices, edge servers and Cloud Computing servers. For such a purpose, a mobile application was developed using Android devices and, for mobile edge servers, the AllJoyn framework [19] was used for device discovery, Peer-to-Peer (P2P) network formation, and data download.

The protocols used by the previously mentioned publications can be compared according to their ability to create opportunistic solutions. For such a purpose, Table 2 indicates the protocols employed by the analyzed state-of-the-art solutions, while Table 3 compares them based on the most relevant opportunistic features, which include:Reliability: it is the capacity to notify the sender if the delivery of data to recipients was successful or not.Packet loss recovery: it is defined as the ability to receive a packet after there are problems during its sending.Routed destination: this feature enables routing the transmitted data packets to the intended receiver.Forward: it is the ability to send packets to another node if the receiving node is not the end receiver.Encryption: it is the process of encrypting data so that it cannot be read by third parties.

As it can be observed in Table 3, physical layer protocols such as IEEE 802.11 (WiFi) and GSM do not fulfill any of the previously mentioned opportunistic features. Regarding UDP and TCP, which are transport layer protocols, only the latter provides reliability thanks to its error control functionality. LoRaWAN is another transport protocol shown in Table 2, which uses AES-128 to secure connections and has a mechanism of Quality of Service that is inbuilt into the MAC layer and that allows for using reception confirmation messages.

With respect to UPnP, it is used for network discovery but it cannot route packets opportunistically to the proper destination. In contrast, RPL, together with 6LoWPAN, allows for routing packets. Regarding the protocols based on blockchain, they are able to recover data from the distributed ledger, in which transactions, once recorded, cannot be modified or deleted, as well as encrypted through the use of hash algorithms such as SHA-256.

With respect to libp2p, the networking protocol stack chosen for the system presented in this article, it fulfills almost all of the requirements: peer routing uses a distributed hash table via the Kademlia routing algorithm, so data forwarding is possible; information is replicated between all peers, and when a peer disconnects, libp2p provides a content routing interface where it does not matter who has it stored because it can be verified its integrity by using public key cryptography. Moreover, libp2p makes use of TLS 1.3 to encrypt communications.

Finally, the article [5] describes a proprietary application layer protocol for edge computing vehicles as an interactive protocol with basic request and response operations for the provision of services. The authors mention that standard cryptographic primitives, such as asymmetric/symmetric key-based encryption and digital signatures, are used, but without clearly detailing the specifically used mechanisms.

As a result of the analysis of the protocols compared in Table 3, the proposed solution makes use of TCP together with libp2p, thus fulfilling all the desired opportunistic features.

### 2.2. Bluetooth 5 Features for OEC Networks

In recent years, Bluetooth has become a widespread technology and has been used in many IoT systems [20]. Specifically, BLE is currently positioned as one of the most energy-efficient technologies for wireless communications [21]. In addition, BLE can be fully decentralized, providing communications between nodes via advertisements through BLE Mesh, which makes it especially attractive for opportunistic communications.

With the introduction of Bluetooth 5 [22], the features of the previous versions have been significantly improved, increasing bandwidth, range and improving coexistence considerably, but BLE Mesh is only compatible with the previous version of Bluetooth (i.e., with Bluetooth 4.x).

The use of Bluetooth 5 with Bluetooth Mesh would allow for improving certain aspects and for supporting new applications. For example, one of the limitations of current BLE Mesh is its throughput, which is limited by aspects like the fact of being implemented as a broadcasting communication (controlled flooding), which makes it necessary to establish an advertisement time (this does not occur in direct communications, which, after establishing the connection, a constant data flow is exchanged). Since only 3 channels are available for broadcasting advertisements, an interval has to be defined to avoid flooding, which reduces throughput considerably. Moreover, the effective payload of an advertisement packet has to be small, so, if a significant amount of data needs to be sent in an advertisement, it is necessary to use segmentation, which further reduces the effective payload.

Fortunately, Bluetooth 5 provides a higher bandwidth to be used with extended advertisements, which are not restricted to the three advertisement channels: they can be transmitted on all the other channels used for data transmission. This enables that a standard that is intended for small data transmissions (e.g., for sending commands or sensor data) can be used for transmitting larger payloads (e.g., for transmitting audio or pictures). Pérez-Díaz-de-Cerio et al. [23] analyzed such a case, concluding that the effective throughput of a single-hop under ideal conditions can be increased from 3.79 kbps to near 500 kbps.

Another improvement provided by Bluetooth 5 is its range. BLE is usually intended for short-range communications, but Bluetooth 5 includes a long-range mode that allows for increasing the sensitivity (and therefore the communications range), which allows for reaching the same theoretical sensitivity as technologies based on IEEE 802.15.4 (e.g., ZigBee, Thread, Ant+) [24]. Such a feature is especially interesting for opportunistic systems that need to provide services over long distances or that work in environments that are ’noisy’ in terms of electro-magnetic interference (e.g., in industrial scenarios [25]. However, it must be noted that Bluetooth 5 long-range mode requires increasing the frame size to include error correction codes (thus reducing the actual bit rate). This implies that frame processing will take longer, which derives into higher power consumption (when comparing transmissions at the same transmit power level) and longer advertisement events, which may end up into a higher saturation of the communication channel. Nonetheless, since advertisement broadcasting has to be performed at a certain time interval, and since payload size is really small when using Bluetooth Mesh, it is possible to adjust the time interval so that the total time for sending a message remains the same. Therefore, better link quality can be achieved by simply compromising on consumption and channel occupancy, with no losses in total response time. This allows the Mesh protocol to be used in relatively long-distance scenarios, thus reducing the number of necessary intermediate devices (i.e., relays) and overcoming the communication distance limitations of BLE.

BLE also has another two limitations:The BLE protocol offers a feature for configuring nodes as cache devices in order to act as opportunistic nodes (these are called *friend nodes*). However, this cache mechanism is not distributed: if a friend node loses connectivity with an IoT node, the cache of this latter node is lost.The BLE protocol is part of the Bluetooth 4.x set of standards, so none of the different modulations provided by Bluetooth 5 version are officially supported, thus impeding the use of the long-range mode, which are useful for many OEC systems.

To overcome the previous limitations, two measures have been taken in the development presented in this article:The friend node feature of BLE is replaced by a P2P implementation of the cache storage.To compensate the lack of support for long-range communications, the BLE Mesh protocol was modified to operate using Bluetooth 5 modulations. However, it is important to consider that the Long Range modulation is not applicable to any OEC application, since it achieves a greater propagation distance at the cost of increasing its time on air, which implies a higher energy consumption and the saturation of the wireless media.

It is worth pointing out that only a few previous works have tried to make use of Bluetooth for OEC systems. For instance, in [9] the authors detail an opportunistic network architecture for wildlife monitoring that is made up of IoT devices carried by animals. The proposed architecture takes advantage of the use of opportunistic mobile networks over Low-Power Wide Area Network (LPWAN) infrastructure. As part of the infrastructure, a LoRa-based network is deployed, which provides a wide communications range and low-power consumption. Making use of such an architecture, the authors investigate the use of existing BLE-based opportunistic data collection protocols.

Only a few academic OEC solutions use Bluetooth 5. For example, in [26] the authors simulate an architecture for healthcare applications that is able to collect data opportunistically from patients through Bluetooth 5. The presented simulation results show that the proposed model is feasible and can provide timely and reliable communication.

### 2.3. Conclusions on the Analysis of the State of the Art

After analyzing the state of the art, it can be concluded that most developments do not implement decentralized systems that are independent from a remote central cloud. Moreover, there is barely any solution in the literature that makes use of Bluetooth 5 for opportunistic communications (WiFi and cellular networks are the most common communications technologies). Furthermore, most of the features that are necessary for implementing opportunistic communications are currently not provided together by any of the analyzed developments. As a consequence, and to tackle these issues, this article proposes a system that brings together all the characteristics indicated in Table 1 and proposes their implementation with simple and low-cost infrastructure.

Specifically, the architecture proposed in this article provides enhanced features with respect to the other solutions analyzed in Table 1:The use of mobile Fog Computing allows for providing Edge Computing functionality, thus decreasing the time response to the IoT nodes through low-cost gateways that can be scattered throughout large environments to provide opportunistic communications.The use of a recent communications technology like Bluetooth 5 enables carrying out mobile data exchanges with a longer range and less energy consumption with respect to other technologies such as WiFi, GSM, 4G, or 5G. In addition, the use of Bluetooth 5 does not involve mobility restrictions (like it happens with Ethernet) or transmission restrictions (as it occurs with LoRa, which is limited in terms of transmission speed and packets per time unit).As it will be detailed later in Section 3, the proposed architecture has been designed to be jointly decentralized, distributed, scalable and modular. Such features have not been found together in any of the analyzed state-of-the-art systems.The devised architecture also includes additional features such as Resource Sharing, Data Routing and Node Discovery, have not been implemented together in the analyzed state-of-the-art systems.

## 3. Design and Implementation of the Proposed OEC System

### 3.1. Communications Architecture

Figure 2 shows the proposed OEC IoT architecture, which is composed of three layers:IoT Network layer. This layer is at the bottom of the architecture and includes the OEC end nodes. As an example, Figure 2 depicts three different IoT networks (A, B and C) that are capable of exchanging data with the upper layer and of sending them to other nodes that belong to the same network (as it is illustrated with the relay node of IoT network B). The IoT OEC devices of this layer make use of sensors to collect information from diverse scattered scenarios and then they send them for processing to the upper layer when it is detected that opportunistic services are available.OEC Smart Gateway Layer. This layer consists of OEC gateways that have the ability to provide opportunistic services to the deployed IoT nodes with reduced latency thanks to the proximity to them. Moreover, smart gateways can collaborate among them in order to perform more complex tasks or to exchange data from different IoT networks without making use of the Cloud Layer.Cloud layer. This layer is responsible for providing services that cannot be provided by the OEC Smart gateway Layer, like the ones involving heavy processing tasks or large data storage.

### 3.2. Bluetooth 5-Based Implemented Architecture

The implementation of the communications architecture previously depicted in Figure 2 is shown in Figure 3. As it can be observed, communications among nodes and between nodes and gateways is carried out through Bluetooth 5. Bluetooth 5 is still being adopted worldwide, but it provides multiple benefits respect to the previous Bluetooth versions in terms of low power consumption and long communications range [27]. Regarding the communications among the OEC smart gateways, they are performed through WiFi/4G networks due to their ease of implementation and deployment.

The two upper layers of the architecture provide different essential services. In the case of the OEC Smart Gateway Layer, it provides peer discovery, peer routing, data routing and resource sharing services. Regarding the Cloud Layer, it provides the routing service that enables communicating between different IoT networks whose gateways are not connected with each other (either directly or through other gateways).

The following subsections describe in detail how the different components are interconnected and how they have been implemented with the objective of easing the development of OEC applications.

### 3.3. End-to-End Latency Model

Taking into consideration the proposed Bluetooth 5-based implemented architecture, the total latency of the devised system (i.e., its end-to-end latency) can be modeled as indicated in Equation (Equation 1) and includes the time needed to connect to the bootstrap node ($t_{bootstrap\_conection}$), the time it takes to discover all the nodes connected to the same network ($t_{connect\_time}$) and the time required to transmit data between nodes ($t_{data\_transmission}$):(1)$$t_{total}=t_{bootstrap\_conection}+t_{connect\_time}+t_{data\_transmission}$$

The last term of Equation (Equation 1) ($t_{data\_transmission}$) can be divided into the three latencies indicated in Equation (Equation 2): the time needed to send the data to a nearby gateway ($t_{sending}$); the time it takes to upload the data to the network ($t_{upload}$) and the time required by the destination node to receive the data ($t_{reception}$):(2)$$t_{data\_transmission}=t_{sending}+t_{upload}+t_{reception}$$

Specifically, the measurement of $t_{data\_transmission}$ requires the following steps:
The data are initially sent from the transmitting node to a smart OEC gateway in the same network via BLE (the required time is which is called $t_{sending}$).Then, the data are received by an OEC gateway through its serial port (to which a Bluetooth 5 development kit was connected) and posted to the DHT network. The time needed for sharing the data through the DHT networks is what is defined as $t_{upload}$.Finally, the OEC gateway that is located in the same opportunistic network as the IoT destination node, collects the sent data from the DHT network and delivers the message to the IoT node over Bluetooth 5 (using BLE). The time required for performing such operations is called $t_{reception}$.

The latencies of Equations (Equation 1) and (Equation 2) are later measured and analyzed in Section 4 for a real environment.

### 3.4. Protocol Stack

The protocol stack proposed to implement the designed architecture consists of six interconnected layers with bidirectional data exchange capacity, as is shown on the left of Figure 4:The Physical (PHY) Layer is responsible for managing the hardware to transmit and receive information via radio waves. For example, in the case of using BLE, data are transmitted using a scheme called Gaussian Frequency-Shift Keying (GFSK).The Medium-Access Control (MAC) Layer has the purpose of managing medium access through OEC devices that implement different PHY layers. Thus, this layer is responsible for identifying the communicating nodes within the coverage area, which can be enhanced by using cost-effective algorithms in deployments with mobile and static nodes [28].The Transport Layer isolates the upper layers from any changes that may occur in terms of hardware in the lower layers (since PHY and MAC layers are usually implemented directly by specific hardware). There are multiple protocols that can be used to support this layer [29,30], such as TCP (Transport Control Protocol), QUIC (Quick UDP Internet Connections) [31], CJDNS [32], UDT (UDP-based Data Transfer Protocol) [33], WEBRTC (Web Real-Time Communications) [34] and UTP (Micro Transport Protocol) [35].The OEC Network Layer provides four services that enable P2P communications:
-Peer discovery and peer routing. When a peer needs to send a message to another peer it first has to know the destination PeerId and its network address. Thus, peer discovery allows for discovering peer addresses thanks to the knowledge provided by the other peers: every contact with a new peer increases the chances of finding the peer that is looked for, while completing the composition of the OEC network, which is reflected in the peer routing tables. The peer routing subsystem exposes an interface to identify the peers to which the message should be routed to. Specifically, the routing subsystem receives a key and returns a list of peers with their information. To perform this task, it is possible to use several technologies, among which Kademlia DHT [36] and multicast DNS (mDNS) [37] are common choices. Nonetheless, other routing mechanisms could be used, such as Domain Name System (DNS) [38], Koorde [39,40] or Chord [41,42]. Similarly, other peer discovery can also be performed through protocols such as DNS, bootstrapping [38] or Peer EXchange (PEX) [43].-Data routing. Data routing uses mechanisms to forward messages to peers if the receiving peer is not in range of the sending peer (this is really common in opportunistic systems). There are several alternatives to implement data routing in OEC systems, such as mDNS, ICE (Interactive Connectivity Establishment) [44], Publish-Subscribe (PubSub) mechanisms and Kademlia DHT. PubSub and Kademlia DHT are two of the most popular, being the former content-based, while the latter is focused on key-based routing:
∗PubSub allows for implementing asynchronous messaging systems. In such systems an IoT node can publish content in the messaging system, which is responsible for sending it to the other peers that are subscribed to it. Thus, messages are delivered to groups of interested peers without depending on a centralized infrastructure. Libp2p provides several PubSub P2P implementations (e.g., gossipsub, floodsub, fpisub) that enable real-time application development. Currently, libp2p uses gossipsub by default, which is called so because peers “talk” to each other about the messages they have detected.∗In the case of Kademlia DHT, to store a key-value pair, the closest *k* nodes to the key are located and the pair is sent to them for storage. In addition, each node forwards it to the rest of the nodes, which ensures the persistence of the pair with a very high probability. Pairs expire 24 hours after publication to limit the existence of outdated index information. As is described in [45], in order to find a pair, a node starts by performing a search to find the *k* nodes with the IDs that are closest to the key. The process stops immediately when any node returns the requested value. For caching purposes, once the lookup is successful, the requesting node stores the pair in the nearest node that did not return the corresponding value.-Resource sharing. It is actually an advanced service that is optional, since not all OEC devices need to make use of it. There are many ways to share resources among IoT devices (e.g., shared bus memories, databases, file sharing protocols...), but, in terms of decentralization, blockchain is one of the most promising [13], since it provides trustworthy and secure communications among parties that have not meet before [46].The OEC Decentralized Data Store Layer is responsible for data structuring and identification, thus providing encryption and authentication.Finally, the Application Layer allows IoT devices to access the exchanged data for their processing and thus implement OEC applications.

### 3.5. Implemented Functionality

The implemented functionality is based on the developed protocol stack, which relies on decentralized technologies such as the ones indicated on the right of Figure 4. Among such technologies, the most critical for implementing OEC systems are arguably the ones related to the OEC Network Layer. For such a purpose, libp2p [47] was chosen, which, thanks to its modularity, can be easily adapted to diverse OEC architectures.

Specifically, libp2p provides a set of protocols for the development of P2P applications. Libp2p started as part of the IPFS project [48] and currently has multiple implementations (e.g., in Go, JavaScript, Rust, Python and C++) that provide flexible solutions for packet transport, security, P2P routing and content discovery. Since libp2p is transport agnostic, it allows OEC developers to select the most appropriate transport protocol for each application. In any case, transport is protected through secure channels that prevent communications between peers to be read by third parties: each peer has a private key, which is kept secret, and a public key that is shared with the other peers. The cryptographic hash of the public key is called PeerId, which is used for identifying every peer unambiguously. To discover the rest of the peers, libp2p makes use of a distributed hash table and of the Kademlia routing algorithm [36] to route requests to the destination PeerId.

Therefore, libp2p is the basis for the main implemented features:Peer discovery and peer routing. For the implementation presented in this article, among the multiple options mentioned in the previous section, mDNS was ruled out because with such a protocol the nodes are only capable of exchanging data within local networks. This was essentially the reason for selecting Kademlia DHT. Such a protocol is based on the use of Distributed Hash Tables (DHT), which can be used to locate items in decentralized P2P networks. The entries of a Kademlia routing table are called contacts and are organized as an unbalanced routing tree. The leaves of such a tree are lists of *k* contacts, called *k-buckets*. Each contact consists of an ID, an IP address and a set of ports. The routing to a specific peer is carried out iteratively: every peer that is on the path toward the destination peer indicates the next hop toward the sending node. This kind of routing is slower than traditional recursive routing strategies, but it is more robust against message losses and simplifies network tracing.Data routing. Both PubSub and Kademlia DHT can be used for implementing OEC systems and it is not straightforward to select one over the other, since there are not detailed analyses on the literature that compare their performance (such a study is out of the scope of this article). In the absence of such a performance comparison, for the work presented in this paper, Kademlia DHT was selected, since it is already used for peer discovery, thus simplifying debugging and reducing development complexity.Security. Security transports are components of libp2p that encrypt information as it is sent over the network. Thus, such information can only be decrypted by the destination peer. Specifically, TLS is used to provide a secure channel between two peers. In 2020, TLS 1.3 [49] became the default security transport in libp2p. TLS 1.3 provides more privacy than TLS 1.2 in data exchanges by making the handshake more secure. This is due to several changes. One of them is the removal of support for many cipher suites, now supporting only 5 [50]. To start the handshake, the client sends the list of supported cipher suites along with the shared key. The server generates the master key with its shared key along with the client’s and responds with an already encrypted message to the client. Finally, the client verifies the server’s certificate and generates the same master key, since it has the server’s shared key. At this point, the communication between the peers can begin. Although the implementation presented in this article makes use of TLS 1.3, it must be noted that TLS 1.3 is not accessible in browser contexts, so not all libp2p implementations can make it the default security transport. The most popular web browsers include support for TLS 1.3, but there is still no way to attach the necessary identity information to libp2p. Due to these problems, the Noise protocol framework was created [51]. Noise allows for composing widely supported cryptographic primitives, Diffie-Hellman key exchange functions, symmetric ciphers and hash functions, which support the different implementations of libp2p [52].

### 3.6. OEC Smart Gateway Firmware

The OEC smart gateway firmware was developed in Go using the libp2p library [53]. The tasks performed by such software are:The creation of a libp2p host. Such a host will create a DHT.Next, the host connects to the bootstrap node, located in the cloud. Such a node is responsible for providing the initial configuration to the joining nodes so that they can join the opportunistic network.The host then performs network discovery and finds all peers that share the same symmetric key. The key (also called rendezvous-point) is a string that is used by the peers to announce their presence.Then, the host iterates over the list of available peers and tries to open a connection with each of them. If a direct connection with a specific peer is not possible, the Cloud will be used as a relay, thus acting as an intermediary between the two communicating nodes.As it will described later in Section 4.1, the firmware needs to open a connection through the serial port, which will be in charge of receiving the data coming from the IoT communication module through the attached Bluetooth module. These data are stored in the DHT and, therefore, will be accessible to the rest of the peers.

## 4. Experiments

This section describes the experiments performed with the developed system. The experimental OEC testbed is first described and then the different tests are detailed. Such tests are focused on measuring the latency of the system, since it is key for an opportunistic system when implementing practical OEC applications (in real scenarios, high latencies impede to implement many IoT applications, especially the ones that require real-time or near real-time data exchanges). All the source code necessary for replicating the presented experiments is available online [54], thus allowing any researcher to use and to extend the developed OEC system.

### 4.1. Experimental Testbed

To evaluate the performance of the proposed OEC system, an experimental testbed was built. The components of such a testbed are depicted in Figure 5 and include two IoT nodes based on a SBC (Raspberry Pi 3B) that has a Nordic nRF52840 development kit connected through the serial port to add Bluetooth Mesh support. One of such IoT nodes is shown in Figure 6. Nodes were flashed with the developed software and then provisioned with the Nordic nRF Mesh app for Android.

Figure 5 also shows the hardware of the deployed OEC gateways, which made use of two different SBCs with the purpose of comparing their performance during the tests. Specifically, a Raspberry Pi 3B+ and a Raspberry Pi Zero were used, whose main characteristics are shown in Table 4. Like for the OEC IoT nodes, the gateways make use of two Nordic nRF52840 development kits for providing Bluetooth Mesh (all Bluetooth development kits were provisioned on the same network to enable their intercommunication).

The scenario where the tests took place was located indoors, in an office. There, the two gateways and the two IoT nodes were placed in areas where they cannot detect each other. Each IoT node was static at roughly one meter from its respective gateway. One of the gateways was connected through WiFi to an optic fiber router, while the other one was also connected through WiFi to the Internet, but through a 4G smartphone that acted as a WiFi access point. Regarding the Cloud, it was located in a remote server whose ping was approximately 50 ms when using the optic fiber router and 120 ms for the 4G smartphone.

In order to illustrate how the testbed works, Figure 7 depicts the steps involved in the transmission of data between two opportunistic IoT nodes that are managed by two distant OEC gateways. Thus, the figure indicates which tasks are executed by every different part of the system and which relationships exist among them.

### 4.2. Latency When Running the Bootstrap Node on the Edge and on the Cloud

The first set of experiments was aimed at analyzing the impact on the latency of a critical component of the developed OEC system: the bootstrap node. In particular, it is interesting to analyze how performance varies depending on the location of bootstrap node. Such a node is really important for DHT networks, since IoT nodes can discover the rest of the network by simply connecting to it.

To measure the impact of the location of the bootstrap node, the time it takes for a node to connect to the rest of the nodes of the same network was measured and then some data was saved in a decentralized way in the DHT. The measured time was divided into three different parts: the time it takes for the peer to connect to the bootstrap node (*bootstrap time*), the time to try to connect to the other nodes (*connect time*), and the time it takes to save the collected value (*put time*). Moreover, two situations were distinguished: when the bootstrap node was run locally (as an Edge Computing server) and when it was executed on the Cloud. In both scenarios, the IoT node was connected through Ethernet (to avoid latency oscillations associated with the use of wireless communications) and using a Raspberry Pi 3B. For each test, 50 iterations were performed and five different situations were measured: when 50, 100, 200, 300, and 600 peers were already connected to the network.

The obtained results are shown in Figure 8, Figure 9, Figure 10, Figure 11 and Figure 12. As it can be observed, for the case when the bootstrap node was running on the edge, latency was significantly lower than when running it on the Cloud. Moreover, the results indicate that the number of peers connected to a bootstrap node impact latency: the higher the number of connected peers, the higher the latency. Furthermore, when mixing the bootstrap node location and the number of connected peers, latency changes significantly. For instance, when running the bootstrap node on the Cloud, the average total time (i.e., the end-to-end latency) for 600 connected peers is roughly 40 s slower than for 50 connected peers, while, when deploying the bootstrap node on the edge, such a time difference is only of 3 s.

Latency differences between running the bootstrap node on the Edge and on the Cloud can be easily spotted through Figure 13, which represents together the mean total times (already depicted in Figure 8, Figure 9, Figure 10, Figure 11 and Figure 12) against the number of connected peers. As can be observed, the mean increases with the number of connected peers, being it clearly higher when executing the bootstrap node on the Cloud.

Finally, put and connection times can also be isolated from the rest of the measurements to observe how the number of peers impact them. Thus, Figure 14 shows that put times for the edge bootstrap node remain constant (between 100 and 150 ms) despite the increase in the number of connected peers. Put time oscillates clearly more (between 200 and 1000 ms) when using the bootstrap node on the Cloud, but it can be stated that the number of peers does not impact it.

In contrast, connection times are clearly affected by the number of connected peers: as is shown in Figure 15, the higher the number of peers connected to the network, the longer it takes to communicate with all of them. Moreover, noticeable differences can be observed when comparing edge and Cloud bootstrap node execution. For example, for the edge bootstrap node, connection time was approximately 0.4 s for 100 peers, while it was roughly 6 s when using the bootstrap node deployed on the Cloud.

### 4.3. IoT OEC Device Performance with Different Hardware

The next set of experiments was carried out to determine how the used hardware impacts the performance of the proposed OEC system. Thus, the tests described in the previous subsection were repeated to compare Raspberry Pi Zero and Raspberry Pi 3B+ performance but when making use of WiFi connectivity (this is for the sake of fairness, since the Raspberry Pi Zero does not have an Ethernet port) and when using an edge bootstrap node. For each test, 50 iterations were performed for a number of peers that oscillated between 50 and 600.

The obtained results are shown in Figure 16, Figure 17, Figure 18, Figure 19 and Figure 20. As it can be observed, the hardware for the IoT OEC devices clearly impacts performance: for every scenario, the times obtained when using the Raspberry Pi Zero are higher. Such a difference becomes clear as the number of connected peers increases. For instance, there is a difference of roughly 2 s in total time (i.e., end-to-end latency) between both tested devices for 50 connected peers, which increases to slightly more than 7 s for 600 peers. Therefore, if latency is critical for an OEC application, hardware needs to be selected carefully. Nonetheless, it is recommended to perform empirical tests with the selected hardware, since, if the devices reach a sufficiently low latency, the use of devices like Raspberry Pi Zero may be good enough and are significantly cheaper than other alternatives (specifically, Raspberry Pi Zero costs approximately a fourth of Raspberry Pi 3 B+).

### 4.4. End-to-End Latency between Nodes in Different Opportunistic Networks

The following set of tests measured the time it takes to send data from an IoT node that belonged to a network, to another node located in a remote and different IoT network. Specifically, the following steps were involved:
The data were first sent by using BLE from the sending node to a smart OEC gateway located in the same network.Then, the gateway received the data through its serial port (where a Bluetooth 5 development kit was attached to) and uploaded them to the DHT network.Next, the OEC gateway that operated in the same opportunistic network where the destination node was located, collected the sent data from the DHT network. To make this communication possible, the role of the Cloud, which acted as a relay, was essential, since it allows for connecting gateways that belong to different IoT networks.Finally, the gateway sent the message to the destination node through Bluetooth 5 (using BLE).

The measured latency for 1000 iterations is shown in Figure 21. The average end-to-end latency was 736 ms (with a minimum of 596 ms and a maximum of 1185 ms), with a variance of 4181 ms. Thus, the obtained results show that the proposed opportunistic network, in the tested scenario, provides relatively low end-to-end latency, being low enough for many potential opportunistic applications.

### 4.5. Security Impact on OEC Communications Latency

The implementation of secure schemes has been traditionally a challenge for resource-constrained IoT devices [55]. As was previously mentioned in Section 4.5, in the case of libp2p, TLS is used by default, which can lower IoT system performance [56].

To determine the impact on latency of using high security mechanisms, it was measured the time it took for a node to connect to the rest of the nodes of the same network and then store a value in a decentralized way in the DHT (as performed in Section 4.2) when enabling/disabling TLS 1.3. In both cases, the used IoT OEC device was a Raspberry Pi 3B+ and used WiFi for communications. For each test, 50 iterations were carried out with between 50 and 600 connected peers.

The obtained results are shown in Figure 22, Figure 23, Figure 24, Figure 25 and Figure 26, which show that, in the selected scenario, security clearly impacts IoT node performance: independently of the number of connected peers, the latency obtained when enabling security is higher than when it is disabled. For instance, for 50 connected peers, latency is increased by roughly 600 ms when using TLS 1.3, while such a difference goes up to 4 s for 600 connected peers. Therefore, although high security is always recommended, future developers will have to design their OEC systems to reach a trade-off between security level and latency.

To spot the previously mentioned differences easier, Figure 27 depicts the mean total time required by the OEC system when using or not security. As it can be observed, the mean total time is clearly higher when using security.

Finally, Figure 28 and Figure 29 show the put and connection times for scenarios when 50 to 600 peers are connected to the bootstrap node and when security was or was not enabled. Specifically, Figure 28 shows that put times with security enabled are constant and independent from the number of peers, remaining between 55 and 60 ms (the same occurs when disabling security, oscillating between 25 and 30 ms). Therefore, the use of security doubles put time, but such a difference is actually low in absolute terms.

In contrast, Figure 29 shows larger differences for connection time. This is especially noticeable when the number of peers increases, since more time is dedicated to applying TLS 1.3 to establish each peer communication. For instance, the connection time difference between using or not security, for 50 connected peers, is approximately 0.6 s, while the difference for 600 peers is around 4 s.

Therefore, the use of the default libp2p security mechanism provides high security, but it is harmful for the latency of the system, so time-critical OEC applications should consider alternative security schemes or just disable security in certain circumstances.

## 5. Key Findings

As it could be concluded in light of the previous section that the proposed OEC system can be influenced by multiple factors (e.g., the location of the bootstrap node, the OEC hardware, the use of security). The following are the most relevant findings that can be drawn from the obtained results, which can be useful for future OEC developers and researchers:The performed experiments showed that the obtained latencies allow for performing fast data exchanges, which enable implementing many opportunistic IoT applications. It is difficult to determine how much faster the developed system is in comparison to the systems analyzed in Section 2, since most of them have not performed latency tests. Specifically, among the analyzed works, only the authors of [26] show results related to latency experiments. In such a paper, Bluetooth 5 is included as communications technology, but the obtained latencies are not empirical (they come from simulations) and the simulation scenario differs significantly from the one evaluated in this paper (the scenarios proposed in [26] are related to a remote health monitoring application for rural areas). Considering such differences, the results presented in [26] show latencies between 4 and 17 h to achieve a message delivery rate of a minimum of 38.7%. Therefore, such results are much higher than the end-to-end latencies obtained in this article, which are between 0.93 s and 2.11 s, but for an indoor short-distance scenario.As was previously mentioned, since most of the reviewed publications did not include latency results, the measurements obtained with the proposed architecture were compared with those of a simplified edge-cloud architecture. In order to perform the comparison, a system was created with the same architecture as the one proposed in Section 3, but without making use of opportunistic algorithms. Thus, in such an architecture an IoT node sends data through Bluetooth 5 to the nearest non-opportunistic gateway, which establishes a P2P connection (using the Cloud as a relay) with a remote gateway that sends the data to the destination IoT node via Bluetooth 5. It is important to note that, in this non-opportunistic architecture, the destination node needs to be always available for receiving data; otherwise, the sent data are lost because the distributed storage of the receiving IoT node is not available. The results of such a comparison are shown in Figure 30. It can be observed that the latency differences between the two compared architectures are small. Specifically, in the case of the opportunistic architecture, the average measured latency is 736 ms, while the non-opportunistic Cloud-based architecture communications have an average latency of 715 ms. Such a 21 ms time difference essentially corresponds to the contribution of $t_{upload}$ to Equation (Equation 2). In any case, the observed time difference is so small that most IoT applications will not be aware of the additional latency, while being able to provide all the benefits of using opportunistic communications.It is also possible to compare the cost of deploying the proposed architecture with respect to other state-of-the-art systems. Specifically, Table 5 shows the total cost of the architectures evaluated in Section 2 when compared to the cost of the architecture proposed in this article. For the sake of fairness, the same scenario was assumed for all architectures for estimating the amount of necessary hardware. Such a scenario consisted in a three-story building with 300 $m^{2}$ floors. Thus, the data of the used hardware were obtained from the experiments described in each article but considering the devised scenario. In Table 5, it can be seen that the overall cost of the proposed system is inexpensive when compared to the architectures described in [7,16,17]. In addition, while the overall cost for the proposed scenario is higher than the one required by the architectures detailed in [9,12], if more floors needed to be covered, the proposed opportunistic architecture would only need to replicate a gateway node (because its range can cover the entire floor), while the other state-of-the-art architectures would need to replicate almost all of their infrastructure on each floor.The mathematical terms of the presented latency model (previously described in Section 3.3), can be analyzed by considering the obtained experimental results and the following conclusions can be extracted:
-$t_{bootstrap\_conection}$. In order to measure this latency, tests were carried out in several scenarios and under different network layers (e.g., edge and cloud), different hardware (e.g., Raspberry Pi 3B, Raspberry Pi Zero) and different security configurations. For each test, 50 iterations were made and five situations were measured with 50, 100, 200, 300, and 600 peers connected to the network. The performed experiments showed a constant latency of roughly 50 ms, independently of the scenario.-$t_{connect\_time}$. For this latency, tests were as for $t_{bootstrap\_conection}$. The performed experiments show that latency depends directly on the number of connected peers. For the edge scenario, latency increases about 0.4 s per 100 peers, while in the cloud scenario such a time increases by 6 s.-$t_{data\_transmission}$. To measure this latency, tests were performed with 1000 messages that were sent sequentially from one node in a network to another in a different network. The average latency obtained for these experiments was 1.18 s.-$t_{upload}$. This latency was measured in the same way as for $t_{bootstrap\_conection}$. The obtained results show a latency that oscillates between 100 and 150 ms for the edge scenario and between 200 and 1000 ms for the cloud scenario. In addition, the performed experiments allowed for concluding that this latency does not depend on the number of connected peers.The main detected challenges, as well as further research areas, are:
-Future developers will have to design their OEC systems to reach a trade-off between security level and latency by disabling certain security features under different circumstances or by devising novel security schemes.-The most critical aspects related to the implementation of practical OEC systems are the ones associated with the OEC Network Layer, which provides services for P2P communications (e.g., peer discovery, peer routing, data routing, and resource sharing). Specifically, the implementation of the proposed architecture relies on libp2p, the modular and extensible P2P network stack used by IPFS. More and more technologies are becoming available for such an implementation, but design trade-offs between the different requirements need to be considered (e.g., Kademlia DHT is robust for routing in terms of message losses and simplifies network tracing, but it is slower than other traditional recursive routing strategies).-When deploying an IoT OEC system, hardware requirements must be carefully analyzed and then empirical tests should be performed in order to select the most appropriate IoT nodes and OEC gateways.-The proposed OEC communications architecture, the used protocols and technologies, as well as the experiments outlined throughout this article, were selected having in mind a specific OEC scenario and set of requirements. Therefore, the obtained latency results should not be directly generalized to every OEC scenario. Nonetheless, the interested OEC system developers and researchers can make use of the provided source code to replicate the experiments in different scenarios and to adapt them to other requirements (e.g., cost, consumption, security level, scalability, mobility).

## 6. Conclusions

This article presented a novel IoT OEC system based on Bluetooth 5 nodes. After reviewing the most relevant state-of-the-art OEC solutions, the proposed decentralized communications architecture was described in terms of components, protocol stack, and functionality. To demonstrate the feasiblity of the developed system for implementing practical applications, its latency was measured. The results show that it is possible to provide end-to-end communications between two different IoT networks with low latency. However, different factors should be considered: the location of the bootstrap node (the use of edge nodes reduces communications latency), the selected OEC hardware (some SBCs are faster but more expensive than other alternatives), and the use of security (TLS 1.3 increases latency but provides high security). As a consequence, this article has not only presented a novel solution, but it has also provided useful guidelines for the developers of future OEC systems.

## Figures and Tables

**Figure 1 sensors-22-08360-f001:**
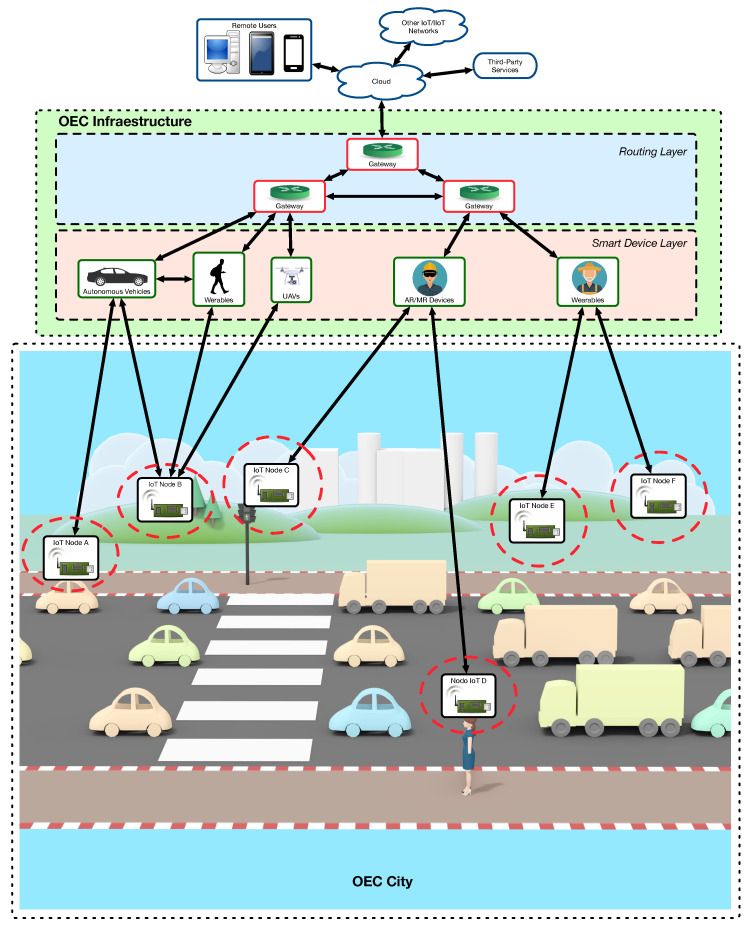
Example of OEC architecture for a smart city.

**Figure 2 sensors-22-08360-f002:**
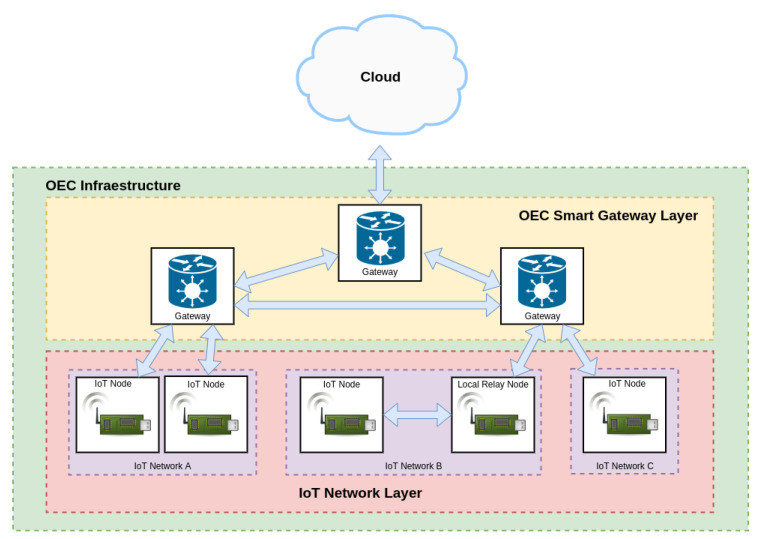
Proposed OEC communications architecture.

**Figure 3 sensors-22-08360-f003:**
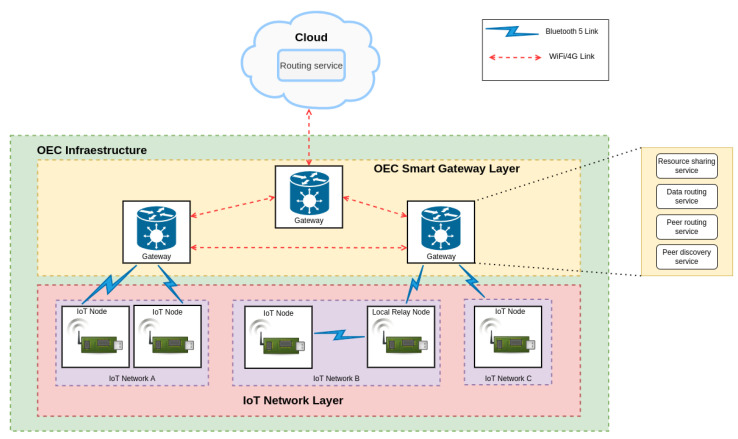
Implemented OEC communications architecture.

**Figure 4 sensors-22-08360-f004:**
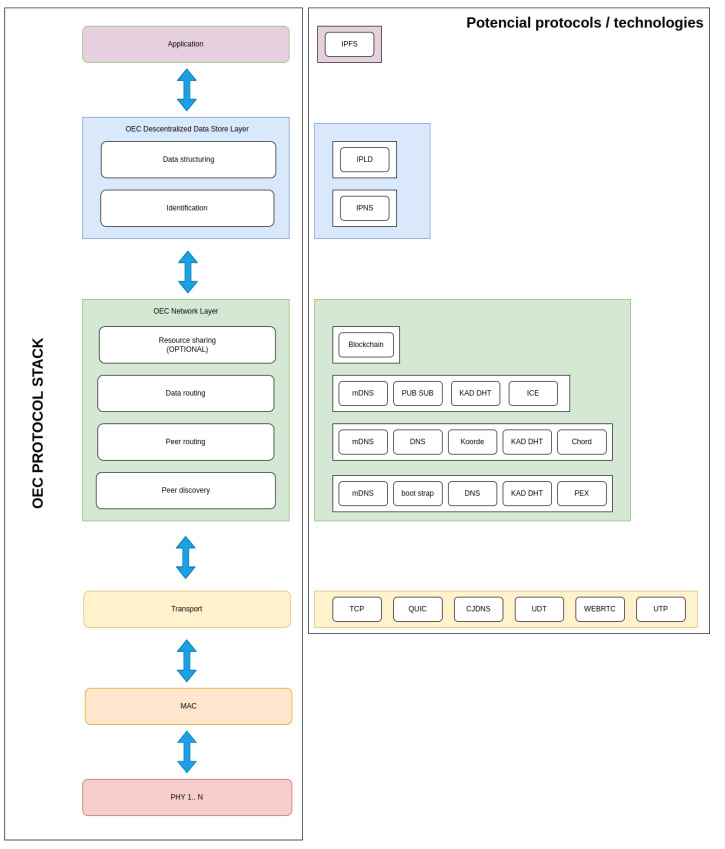
Proposed OEC protocol stack.

**Figure 5 sensors-22-08360-f005:**
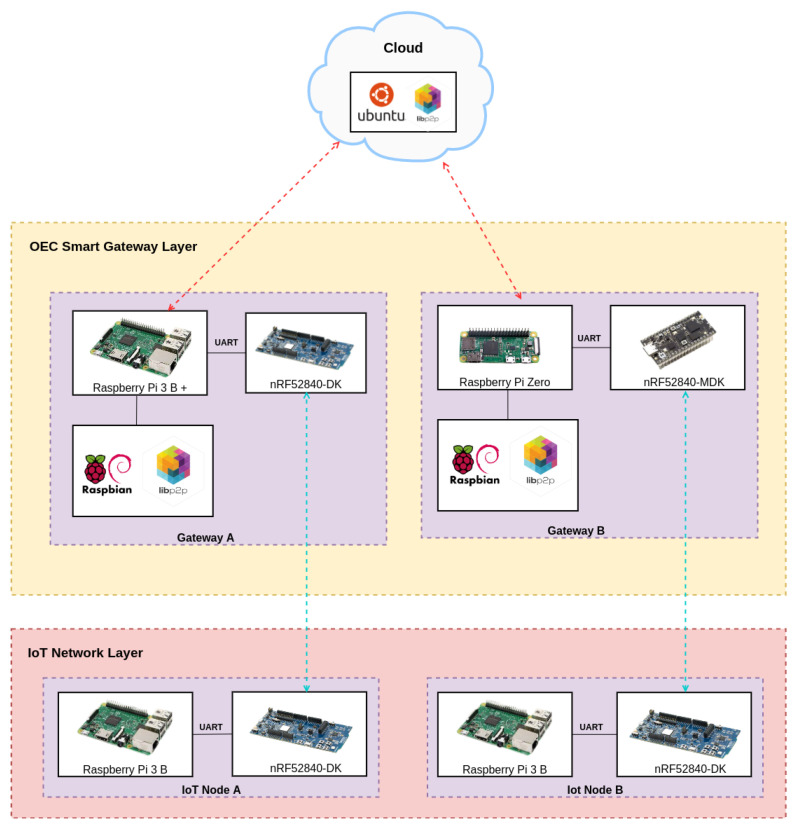
Experimental testbed.

**Figure 6 sensors-22-08360-f006:**
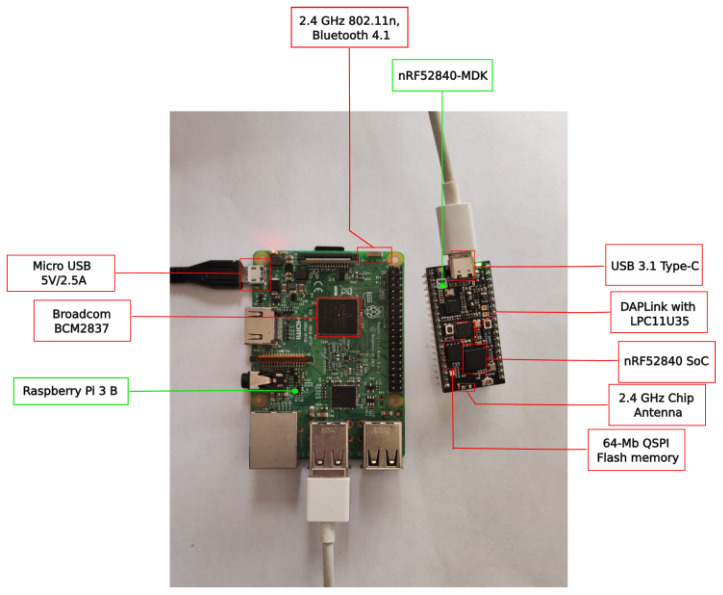
Components of one of the testbed IoT nodes.

**Figure 7 sensors-22-08360-f007:**
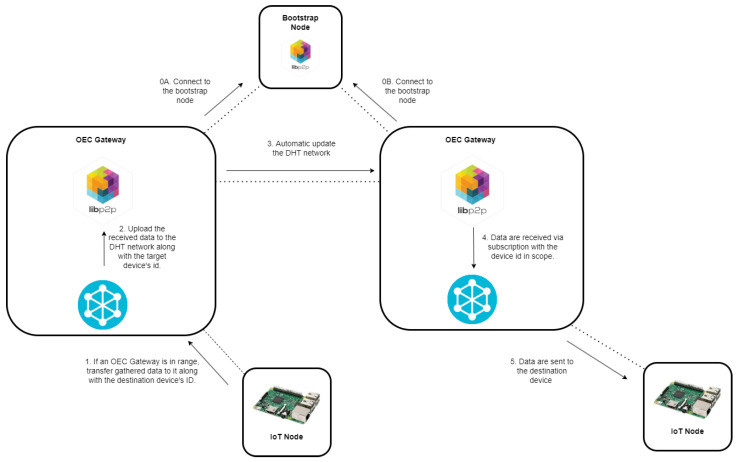
Step-by-step process for exchanging data between two opportunistic IoT nodes.

**Figure 8 sensors-22-08360-f008:**
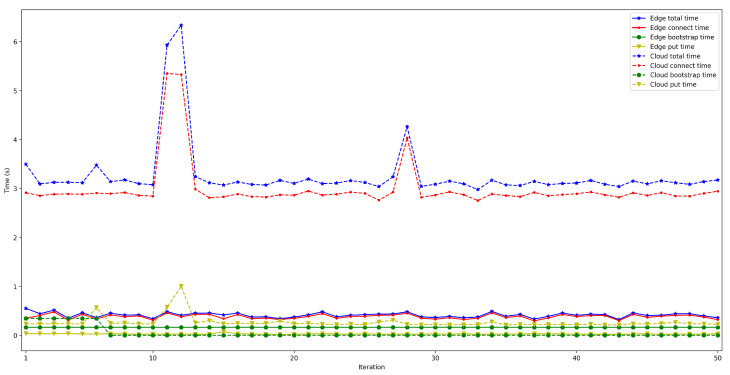
Latency when running the bootstrap node in the edge/Cloud for 50 connected peers.

**Figure 9 sensors-22-08360-f009:**
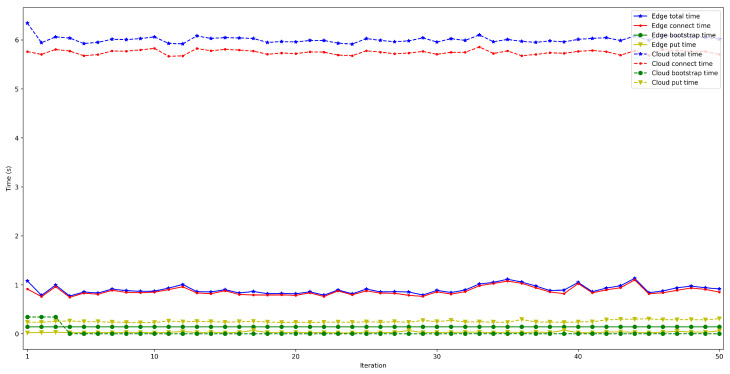
Latency when running the bootstrap node in the edge/Cloud for 100 connected peers.

**Figure 10 sensors-22-08360-f010:**
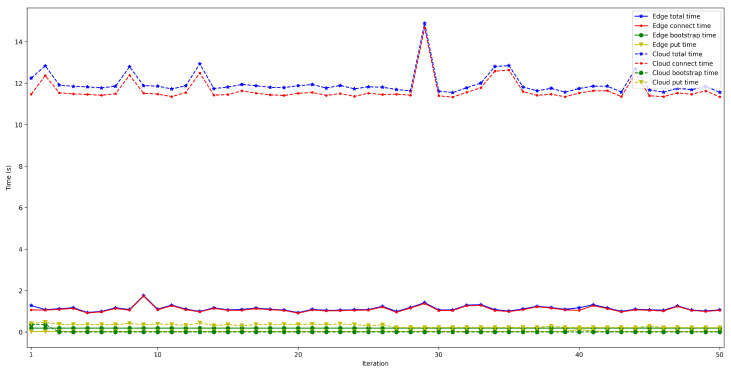
Latency when running the bootstrap node in the edge/Cloud for 200 connected peers.

**Figure 11 sensors-22-08360-f011:**
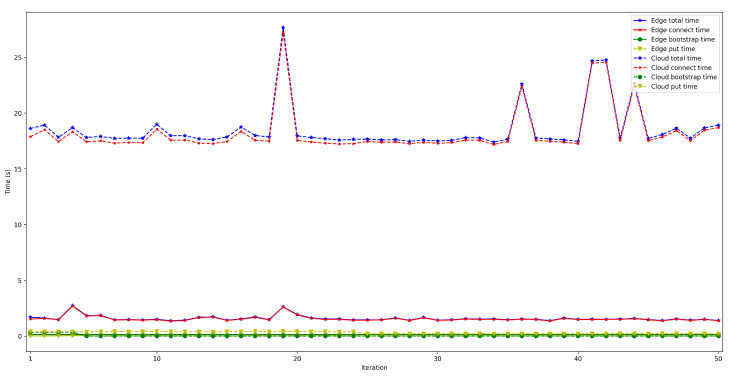
Latency when running the bootstrap node in the edge/Cloud for 300 connected peers.

**Figure 12 sensors-22-08360-f012:**
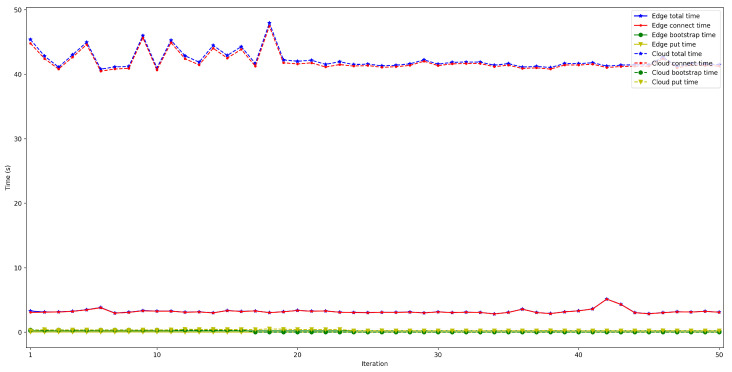
Latency when running the bootstrap node in the edge/cloud for 600 connected peers.

**Figure 13 sensors-22-08360-f013:**
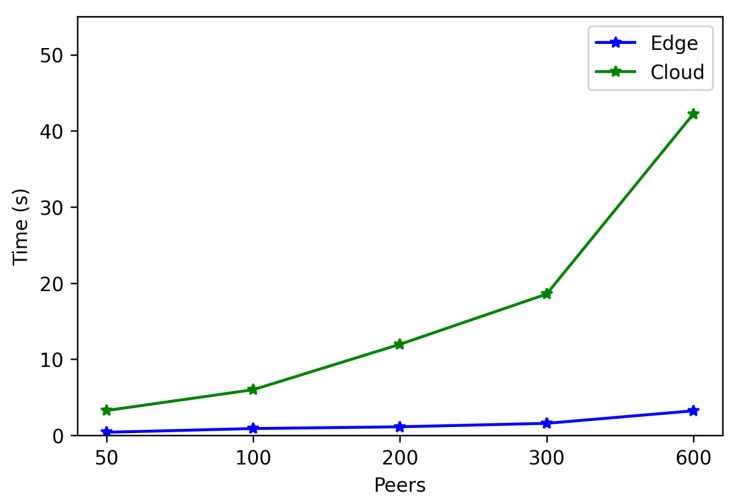
Mean total time when running the bootstrap node in the edge/Cloud.

**Figure 14 sensors-22-08360-f014:**
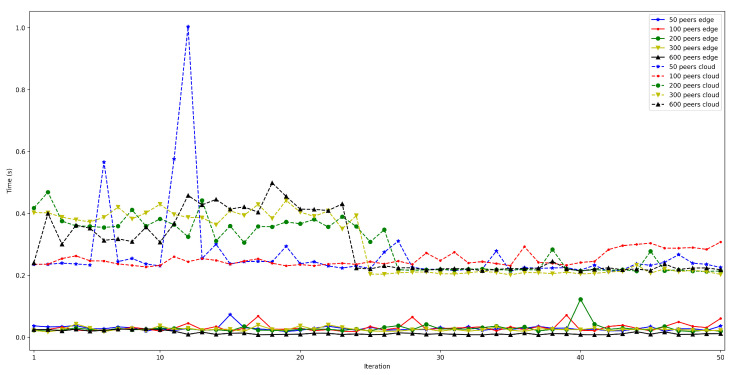
Put time when running the bootstrap node in the edge/cloud.

**Figure 15 sensors-22-08360-f015:**
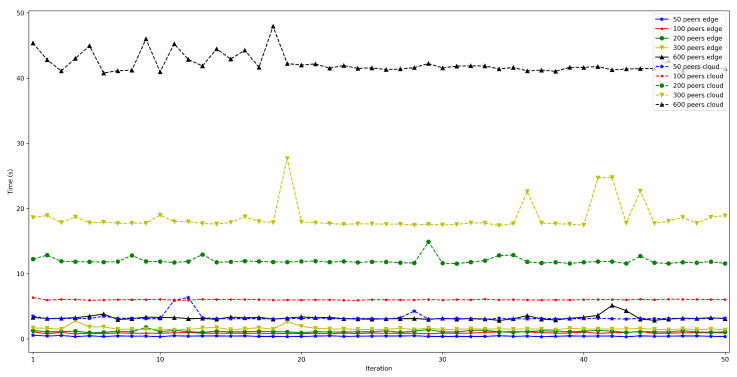
Peer connection time when running the bootstrap node in the edge/cloud.

**Figure 16 sensors-22-08360-f016:**
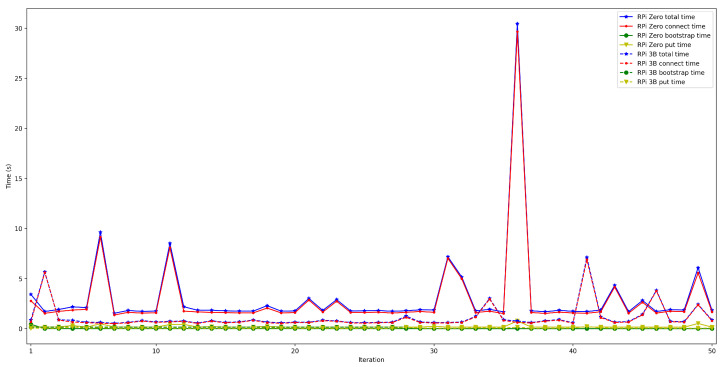
Latency when using Raspberry Pi Zero/3B+ gateways for 50 peers connected to the bootstrap node.

**Figure 17 sensors-22-08360-f017:**
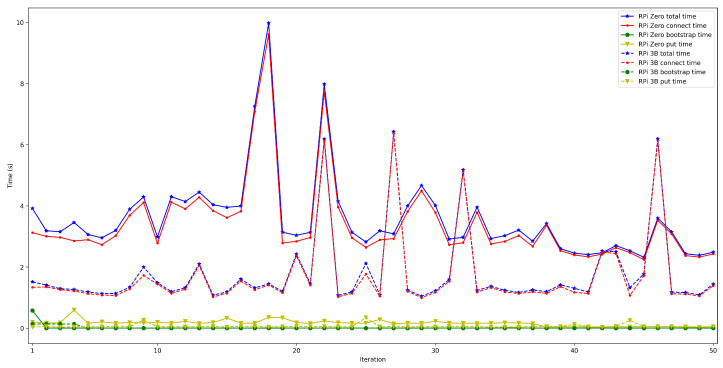
Latency when using Raspberry Pi Zero/3B+ gateways for 100 peers connected to the bootstrap node.

**Figure 18 sensors-22-08360-f018:**
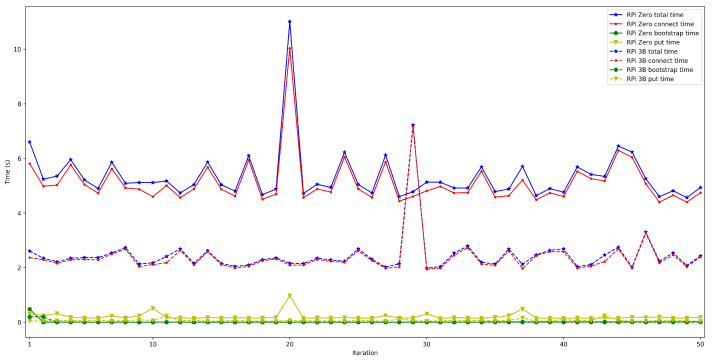
Latency when using Raspberry Pi Zero/3B+ gateways for 200 peers connected to the bootstrap node.

**Figure 19 sensors-22-08360-f019:**
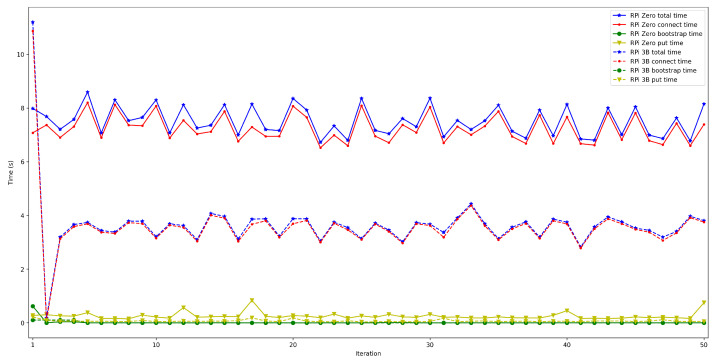
Latency when using Raspberry Pi Zero/3B+ gateways for 300 peers connected to the bootstrap node.

**Figure 20 sensors-22-08360-f020:**
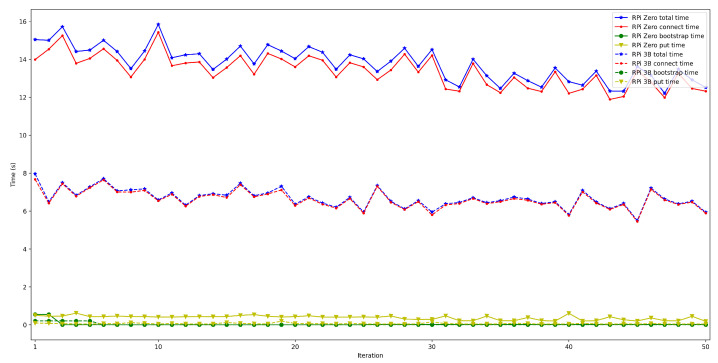
Latency when using Raspberry Pi Zero/3B+ gateways for 600 peers connected to the bootstrap node.

**Figure 21 sensors-22-08360-f021:**
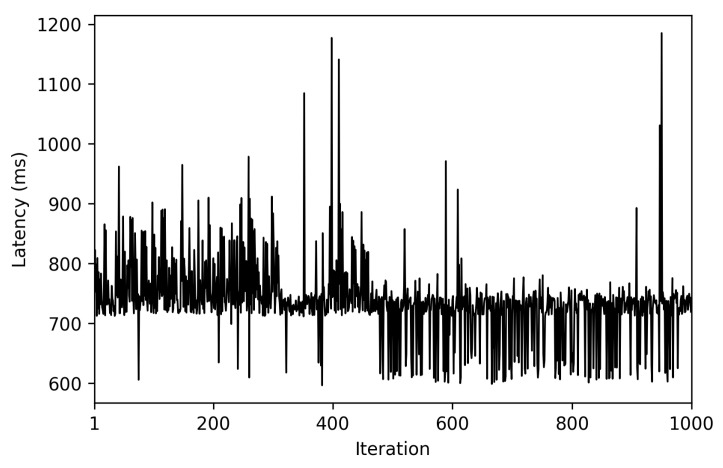
End-to-end latency for 1000 data exchanges.

**Figure 22 sensors-22-08360-f022:**
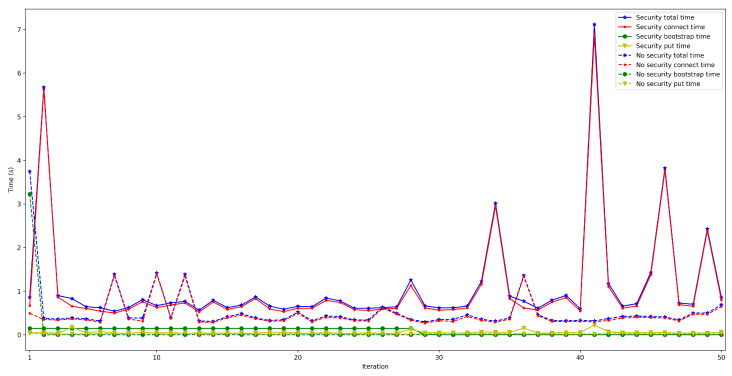
Latency when enabling/disabling TLS 1.3 for 50 peers connected to the bootstrap node.

**Figure 23 sensors-22-08360-f023:**
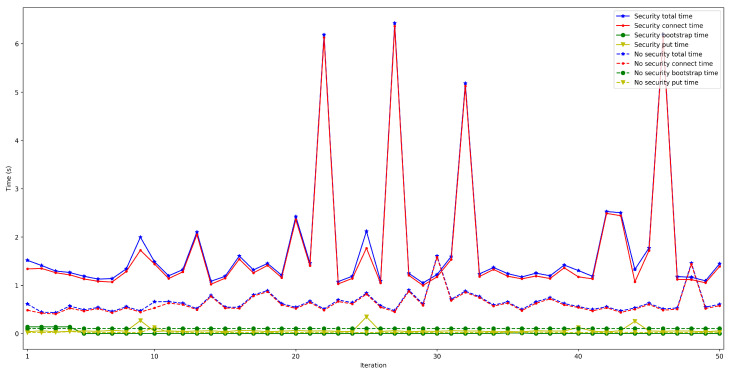
Latency when enabling/disabling TLS 1.3 for 100 peers connected to the bootstrap node.

**Figure 24 sensors-22-08360-f024:**
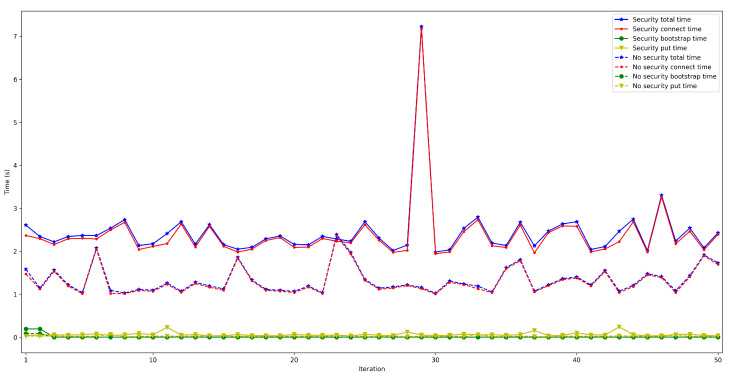
Latency when enabling/disabling TLS 1.3 for 200 peers connected to the bootstrap node.

**Figure 25 sensors-22-08360-f025:**
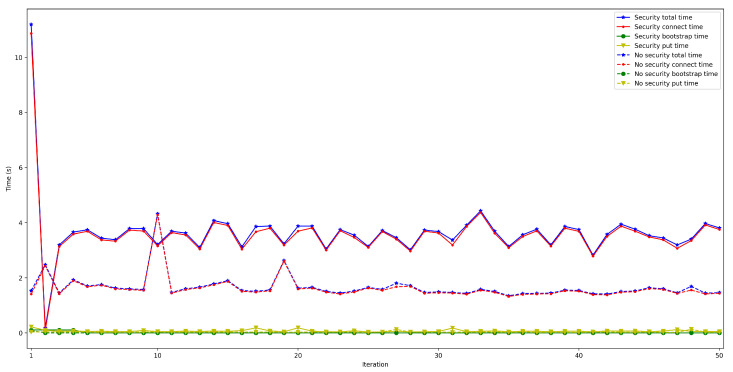
Latency when enabling/disabling TLS 1.3 for 300 peers connected to the bootstrap node.

**Figure 26 sensors-22-08360-f026:**
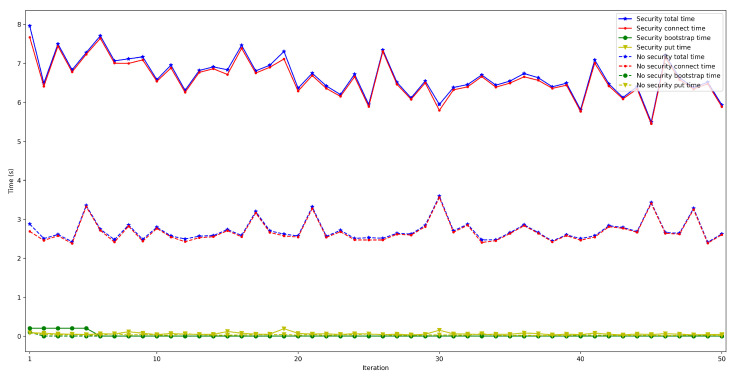
Latency when enabling/disabling TLS 1.3 for 600 peers connected to the bootstrap node.

**Figure 27 sensors-22-08360-f027:**
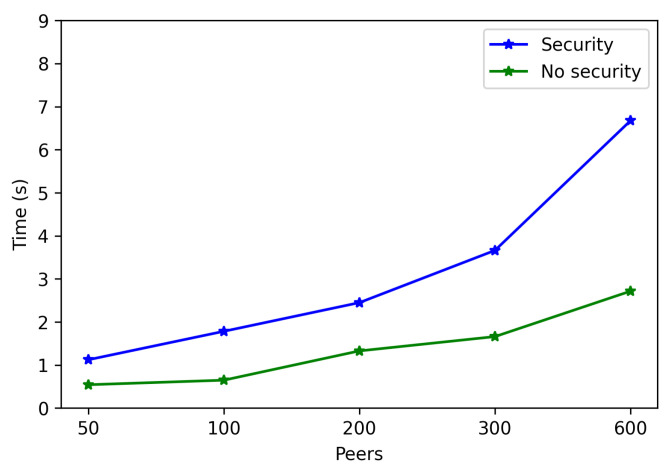
Mean latency when enabling/disabling TLS 1.3.

**Figure 28 sensors-22-08360-f028:**
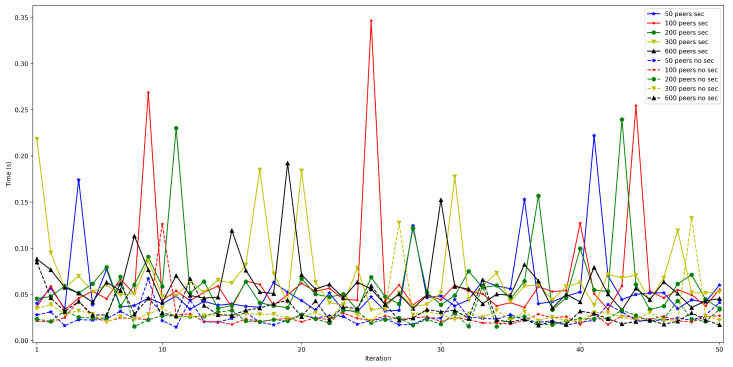
Put time when enabling/disabling TLS 1.3.

**Figure 29 sensors-22-08360-f029:**
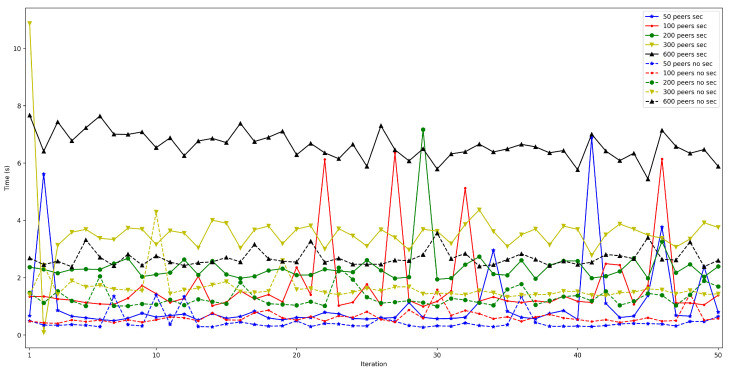
Peer connection time when enabling/disabling TLS 1.3.

**Figure 30 sensors-22-08360-f030:**
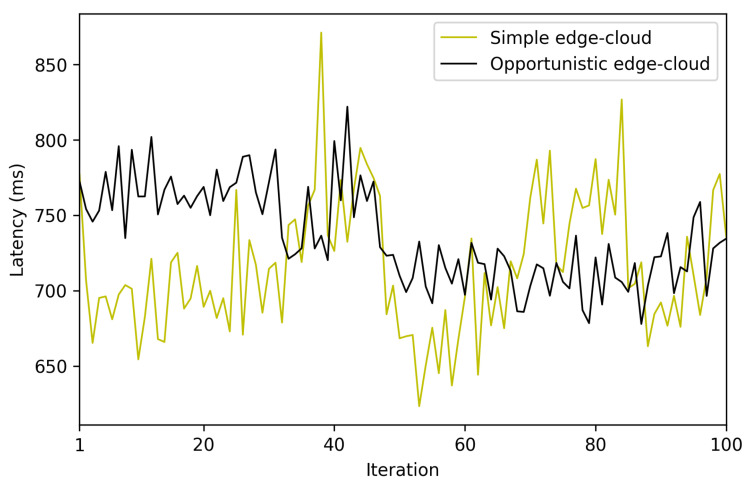
End-to-end latency comparison.

**Table 1 sensors-22-08360-t001:** Main characteristics of the most relevant theoretical and practical OEC solutions.

System	Architecture Type	Application Field	Decentralized	Communication Technologies	Security Mechanisms	Distributed	Scalable	Modular	Resource Sharing	Data Routing	Node Discovery	Hardware	Other Features
[5] (Theoretical Article)	Edge Computing	Parked Vehicles	No	-	Asymmetric and symmetric key-based encryption and digital signatures	Yes	Yes	-	Yes	No	No	-	Parked vehicles act as Edge Computing nodes
[6]	Mobile Edge Computing	-	No	Wi-Fi/GSM	-	Yes	Yes	No	No	No	Yes	Android device	Minimize latency, lower dependencies on Internet connectivity, and reduce the cost of using cloud services
[7]	Blockchain	-	Yes	Ethernet	SHA-256	No	No	No	Yes	No	No	Raspberry Pi 3, Dell 5100	The obtained latency and resource use metrics are similar to the ones for traditional centralized edge approaches
[8] (Theoretical Article)	Fog Computing	-	Yes	-	Server Virtualization	Yes	Yes	Yes	No	No	No	-	High number of potentially connected neighbors and small number of hops
[16]	Fog Computing	UAVs	No	WiFi	-	Yes	Yes	No	No	Yes	Yes	Adafruit CC3000	Low latency services, location aware services, better mobility and access control, better Quality of Service (QoS), more efficient communications
[12]	Fog Computing	-	Yes	-	SHA-256	Yes	No	Yes	No	No	No	Raspberry Pi 4 Model B	Fault-tolerant, secure and auditable
[14] (Theoretical Article)	Edge-Fog-Cloud	Agriculture	No	NB-IoT, WiFi, Zigbee, 5G	-	Yes	Yes	Yes	No	No	No	-	Reduction of energy consumption, CO_2_ emissions and network traffic
[17]	Blockchain Cloud	-	No	-	FS-OpenSecurity (proprietary development [18])	Yes	Yes	No	No	Yes	No	Intel i5 16 GB DDR3-RAM Laptop	High availability, real-time data delivery, high scalability, security, resiliency, and low latency
[9]	Traditional LPWAN (Low-Power Wide Area Network) Architecture	Wildlife monitoring	No	LoRa	-	Yes	Yes	No	No	Yes	Yes	Adafruit Feather 32u4 RFM95 - 868 MHz	High delivery ratio, low latency, low energy
Solution Described in this Article	Mobile Fog Computing	-	Yes	Bluetooth 5, WIFI, 4G	TLS 1.3	Yes	Yes	Yes	Yes	Yes	Yes	Raspberry Pi 3 Model B, Raspberry Pi 3 Model B+, Raspberry Pi Zero	Low cost solution, low latency, SBC-based gateways, for resource-constraint IoT nodes

**Table 2 sensors-22-08360-t002:** Protocols used in theoretical and practical OEC solutions of the state of the art.

Referenced System	Physical Layer	Transport Layer	Network Layer	Application Layer
[5] (Theoretical Article)	-	-	-	Vehicular-edge computing proprietary protocol
[6]	802.11/GSM	-	-	-
[7]	-	TCP/IP	Blockchain-based proprietary protocols	-
[8] (Theoretical Article)	-	UDP	6LowPAN & RPL	-
[9]	-	LoRaWAN	-	-
[12]	-	TCP/IP	Ethereum	-
[14] (Theoretical Article)	-	-	-	-
[16]	802.11	TCP/IP	UPnP	-
[17]	-	TCP/IP	Blockchain-based proprietary protocols	-
System described in this article	-	TCP/IP	LibP2P	-

**Table 3 sensors-22-08360-t003:** Relevant opportunistic features of the protocols of the analyzed state-of-the-art solutions.

Layer	Protocol	Reliability	Packet Loss Recovery	Routed Destination	Forwarding	Encryption
Physical	802.11/GSM	×	×	×	×	×
Transport	TCP	✓	×	×	×	×
Transport	UDP	×	×	×	×	×
Transport	LoRaWAN	✓	×	✓	×	✓
Network	Blockchain-based proprietary protocols	×	✓	✓	×	✓
Network	6LowPAN/RPL	×	×	✓	✓	×
Network	UPnP	×	×	×	×	×
Network	Ethereum	×	✓	✓	×	✓
Network	LibP2P	×	✓	✓	✓	✓
Application	Vehicle Edge Computing Proprietary protocol	×	×	×	×	✓
-	TCP + LibP2P	✓	✓	✓	✓	✓

**Table 4 sensors-22-08360-t004:** Main characteristics of the tested smart gateway hardware.

	Raspberry Pi 3 B	Raspberry Pi 3 B+	Raspberry Pi Zero W
SoC	64-bit ARM Cortex-A53 Quad-Core	64-bit ARM Cortex-A53 Quad-Core	32-bit ARM1176JZF-S Single-Core
Core	Broadcom BCM2837	Broadcom BCM2837	Broadcom BCM2835
GPU	Broadcom VideoCore IV	Broadcom Videocore-IV	Broadcom VideoCore IV
CPU Clock	1.2 GHz	1.4 GHz	1 GHz
Memory	1 GB	1 GB	512 MB
SPI/12C	Yes	Yes	Yes
Supported OS	Linux, Android Things, Windows 10 IoT Core	Linux, Android Things, Windows 10 IoT Core	Linux
Ethernet	10/100 Mbit/s	10/100/1000 Mbit/s	None
Wi-Fi/ Bluetooth	2.4 GHz 802.11n/ Bluetooth 4.1	2.4/5 GHz 802.11ac/ Bluetooth 4.2	2.4 GHz 802.11n/ Bluetooth 4.1
Current Consumption	350 mA	400 mA	160 mA
Cost	USD38	USD45	USD10

**Table 5 sensors-22-08360-t005:** Hardware cost comparison of the proposed solution with other relevant state-of-the-art opportunistic architectures.

Project	Hardware	Cost (€)	#Units	Total (€)
Solution Described in this Article	Raspberry pi Zero W	10.44	3	31.32
	Raspberry Pi 3 B+	51.43	3	154.29
	NRF52840-DK	49	6	294
	PC cloud	600	1	600
				**1079.61**
[9]	Adafruit Feather 32u4 RFM95	35.28	3	105.84
	LARANK 8	393.25	1	393.25
	Adafruit Feather nRF52832	25.95	9	233.55
				**732.64**
[16]	Arduino board+Adafruit CC3000 Wi-Fi board	48.95	1	48.95
	PC UAVFOG	600	2	1200
	PC cloud	600	1	600
				**1848.95**
[7]	Raspberry Pi 3 A+	30.25	9	272.25
	Dell EMC Edge Gateway 5200	3.412.94	1	3412.94
				**3685.19**
[12]	Raspberry pi 4	60	9	540
				**540**
[17]	Cloud Server (64 GB RAM and Intel i7)	640/year	6	3840
	Laptop (16 GB RAM and Intel i5)	600	3	1800
				**5640**

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
