# Peer review of "Practical Latency Analysis of a Bluetooth 5 Decentralized IoT Opportunistic Edge Computing System for Low-Cost SBCs"

_sensors, 2022, doi:10.3390/s22218360_

Round 1

Reviewer 1 Report (Previous Reviewer 2)

Great improvements. I still have a few concerns before it can be accepted to publish. Some of them are very specific to update the old and irrelevant references such as [43] with following recommended one.

Ashraf, S., Alfandi, O., Ahmad, A., Khattak, A. M., Hayat, B., Kim, K. H., & Ullah, A. (2020). Bodacious-Instance Coverage Mechanism for Wireless Sensor Network. In F. Ullah (Ed.), Wireless Communications and Mobile Computing (Vol. 2020, pp. 1–11). Hindawi Limited. https://doi.org/10.1155/2020/8833767 

Author Response

The authors would like to thank the reviewer for his/her valuable comments, which
have certainly helped us to improve the manuscript. Please find attached our detailed responses to the comments. In order to ease the labor of the reviewers we have colored in red the differences with the previous version of the article.

Reviewer 2 Report (Previous Reviewer 1)

The authors have addressed all my concerns. This paper can be accepted.

Author Response

The authors would like to thank the reviewer for his/her valuable comments, which
have certainly helped us to improve the manuscript. 

This manuscript is a resubmission of an earlier submission. The following is a list of the peer review reports and author responses from that submission.

Round 1

Reviewer 1 Report

1.Since there are many existing solutions of decentralized oec (can be seen in table 1), the authors should demonstrate why their proposed architecture is better than existing ones?

2.The main drawback of the paper is that the theoretical basis is weak and no any formula of mathematical model to explain their architecture and network model. The authors should present and explain the network model mathematical.

3.Also, since this paper is a simulation-based testbed, it is required to provide source code or demo, or any additional sources that demonstrate the output of the results (e.g., screenshot, codes etc.)

Author Response

The authors would like to thank the reviewer for his/her valuable comments, which have certainly helped us to improve the manuscript. Please find attached our detailed responses to the comments. In order to ease the labor of the reviewers we have colored in red the differences with the previous version of the article.

Reviewer 2 Report

The missing literature review section and some literature have been mixed with the introduction section. I recommend first re-draft the introduction section according to the following directions.

At the beginning of the introduction section, the authors must emphasize the context of the proposed idea and state the objective either in the form of points or a separate paragraph.

Figure 7. Components of one of the OEC smart gateways is entirely irrelevant here...

Go for a thorough proofread of the paper to rectify several existing typos and grammatical mistakes to improve the written quality of the paper. If necessary take the help of a native English speaker to improve the language of the paper.

The contributions given are unprofessional. The authors should highlight what are the challenges and why they are difficult specifically for the environment in which experiments are being performed. How did the authors solve the problems?

The critical analysis and shortcomings of each existing protocol are missing. Please mention what are the limitation of the current work that needs to be addressed. Secondly, engage the reader and use convincing argument, how the limitation due to congestion and reliability are the prime cause of existing contributions.

References are at satisfactory level.

Author Response

(The authors gave the same response as above.)

Round 2

Reviewer 1 Report

I still find some issues related to the theorical background. For instance, the total latency (Equation 1) which is the main metric used to show the performance of their architecture is not well defined. For instance, how to calculate the data transmission time?  Is it generated, how it is obtained ? how it is measured ?  The mathematical approaches to explain the theorical background is still not adequate, therefore, the paper does not contain the basic background. 

Although, the authors have provided a link to the source code, this link is a general tool for implementation of Bluetooth mesh. Therefore, the authors should explain how the developed and implemented their proposed architecture in the provided link.

Since, this is a testbed-based paper, I advise the authors to clearly explain how all related times are obtained, how the tasks (nodes) are executed in edge or cloud in their architecture. 

The main issue of the paper, is that the authors didn’t compared their architecture with the existing approaches. Their main reason is that it is hard to reproduce existing approach since they have not compared the latency. This reason is not valid since latency is just a metric. Moreover, they can compare their approach (architecture) with a simple edge-cloud architecture scenario or edge-fog-cloud scenario, which is possible. They authors can even add another metric such as the device (IoT or node) runtime in each architecture and cost (money cost) of the architecture.

Without a comprehensible comparison with the existing architectures, this paper lack of novel contributions.

Reviewer 2 Report

In my previous recommendations, I was asked to separate the introduction section from the literature review section but the authors I think couldn't understand my viewpoint and replied as

"we have moved some content to the state of the art (Section 2.1) and we have rewritten the Introduction to make the context and the objectives clear for the reader." Indeed, they didn't follow the recommendations.

I asked in point (5) to mention the shortcomings of previous work but the authors didn't get this point and responded with the irrelevant script as

"To address the concerns of the reviewer, we have

extended Section 3.4 by citing the most relevant protocols for each of the different

layers of the protocol stack. In addition, Section 3.5 has been modified to

state clearly which protocols were used for the implementation of the proposed OEC

system and the reasons behind their selection."